# In vivo noninvasive systemic myography of acute systemic vasoactivity in female pregnant mice

Kristie Huda[1], Dylan J. Lawrence[1,2], Weylan Thompson[2], Sarah H. Lindsey [3] & Carolyn L. Bayer [1] ✉

Altered vasoactivity is a major characteristic of cardiovascular and oncological diseases, and many therapies are therefore targeted to the vasculature. Therapeutics which are selective for the diseased vasculature are ideal, but whole-body selectivity of a therapeutic is challenging to assess in practice. Vessel myography is used to determine the functional mechanisms and evaluate pharmacological responses of vascularly-targeted therapeutics. However, myography can only be performed on ex vivo sections of individual arteries. We have developed methods for implementation of spherical-view photo-acoustic tomography for non-invasive and in vivo myography. Using photoacoustic tomography, we demonstrate the measurement of acute vascular reactivity in the systemic vasculature and the placenta of female pregnant mice in response to two vasodilators. Photoacoustic tomography simultaneously captures the significant acute vasodilation of major arteries and detects selective vasoactivity of the maternal-fetal vasculature. Photoacoustic tomography has the potential to provide invaluable preclinical information on vascular response that cannot be obtained by other established methods.

Vascular reactivity is critical for systemic regulation of systemic blood pressure and blood flow. Measurement of vasoactivity ex vivo provides information about endothelial and smooth muscle cell function and allows dissection of signaling pathways that impact vascular responses. Altered vasoactivity is a major characteristic of cardiometabolic diseases including hypertension[1], diabetes[2,3], atherosclerosis[4], coronary artery disease[5], and renal diseases[6]. Preclinical measurement of vasoactivity is an effective method for determining the efficacy of therapeutics in development for the treatment of cardiometabolic diseases. Vasodilators which are regionally selective would be advantageous for maximizing treatment efficacy while avoiding side effects, such as systemic hypotension. To assess selective vessel responses, wire myography has been the gold standard for assessing and quantifying vascular reactivity. However, myography can only be performed ex vivo on larger, minimally branched vessels[7–10]. As a result, these techniques may not accurately mimic in vivo physiological conditions[11]. Intravital microscopy offers analysis of vascular structure and hemodynamic parameters in vivo, however it requires surgically exposed tissues or implanted optical window for microscopic imaging[12]. Noninvasive ultrasound imaging can be used for measuring in vivo vasodilation[13]. However, standard ultrasound B-mode acquires a 2D field of view, which again limits the vasoactivity measurement to single vessels and doesn't capture the actual physiological changes in 3D space. 3D ultrasound tomography has lower vascular contrast in comparison to photoacoustic imaging[14–16].

Photoacoustic imaging uses nanosecond laser pulses, which can be wavelength-tuned to the optical absorption of hemoglobin. The absorption of light by hemoglobin results in transient heating and expansion in the tissue, followed by cooling and contraction, generating acoustic waves. The resulting photoacoustic signal can be detected using a broadband acoustic array. Photoacoustic imaging has been recently applied to monitor vasoactivity of mouse brains and

[1]Department of Biomedical Engineering, Tulane University, New Orleans, LA, USA. [2]Photosound Technologies Inc., Houston, TX, USA. [3]Department of Pharmacology, Tulane University School of Medicine, New Orleans, LA, USA. ✉e-mail: carolynb@tulane.edu

tumors[17,18]. However, these studies used photoacoustic microscopy systems with slow temporal resolution, limited field of view and limited imaging depth. The spherical-view photoacoustic tomography system used here provides in vivo high resolution anatomical and functional visualization of small animals. Currently, a temporal resolution of 2–10 s is the fastest reported imaging speed for spherical-view geometry[19]. Though this system has a high temporal resolution, the spatial resolution of the system is 390 μm (lateral) and 370 μm (axial) which is larger than the diameter of mouse arteries[20]. Another spherical-view system has a reported spatial resolution of 200 μm (lateral and axial) and large FOV (80 mm)[21]. However, the limited view aperture of the system requires a helical scanning scheme for whole-body imaging, reducing the temporal resolution to 45 s. The TriTom spherical-view photoacoustic tomography system uses a 360° rotation to capture a 3 cm³ volume with a spatial resolution of $173 \pm 17$ μm (transverse plane) that can image vessel diameters down to ~180 mm in 38 s[22].

The placenta, a highly vascularized and dynamic organ, develops during pregnancy to support rapid fetal growth and development. Regulation of the maternal systemic and placental vasculature is essential to meet the high demand for oxygen and nutrients for the fetus. Dysregulated vasoactivity is involved in the major complications of pregnancy including preeclampsia, gestational diabetes mellitus, and fetal growth restriction[23–25]. In particular, vasodilatory pharmaceuticals have been investigated as therapeutics for preeclampsia, a hypertensive disorder of pregnancy. Sildenafil, a selective phosphodiesterase 5 (PDE5) inhibitor widely used for treating pulmonary hypertension[26], and erectile dysfunction[27], was demonstrated preclinically to improve placental function[28–30]. However, off-target effects led to adverse outcomes in infant mortality in clinical trials[31–33]. Assessing the regional responses of vasoactive therapeutics is therefore a critical preclinical step in determining the efficacy and safety of therapeutics intended for use during pregnancy.

We sought to implement spherical-view photoacoustic tomography to assess vascular reactivity both systemically and within the placenta during pregnancy. Studies assessed the systemic vasoactivity of sildenafil along with G-1, a selective agonist of the G protein-coupled estrogen receptor (GPER). Sildenafil induces vascular smooth muscle relaxation via the nitric oxide−soluble guanylate cyclase−cyclic guanosine monophosphate (NO−sGC−cGMP) pathway, inhibiting degradation of cGMP by blocking PDE5. Human chorionic plate arteries[34] and rodent pulmonary arteries are selectively responsive to sildenafil[35].

G-1 has a sex-dependent effect on endothelium-dependent and independent vasodilation in most systemic arteries including uterine arteries[36–39]. Whether pregnancy changes these responses has not been determined.

## Results

### Photoacoustic tomography of systemic in vivo vascular responses

We developed methods to demonstrate the ability of spherical-view photoacoustic tomography to noninvasively measure acute vasodilators in vivo. Figure 1 shows a schematic diagram of the system used to collect 3D images every 38 s of a 3 cm³ volume of the abdomen of pregnant CD-1 mice using a spherical-view photoacoustic tomography system TriTom (PhotoSound Technologies Inc., Houston, TX, USA) integrated with tunable OPO laser (Opotek Phocus Benchtop laser, 5−7 ns pulses, 10 Hz repetition rate; Opotek Inc., Carlsbad, CA, USA). Photoacoustic tomography images were acquired and processed to determine the extent of vasodilation and response time of the major arteries of pregnant CD-1 mice to the vasodilators. Standard modified back-projection algorithm provided by the manufacturer was used for image reconstruction[40]. After reconstruction, the images were normalized by average radio frequency (RF) signal and laser energy measured during imaging. We imaged the whole abdomen of a pregnant mouse in vivo capturing the systemic and fetal vasculature and placentas as shown in Fig. 1b–g.

Figure 2a shows representative 3D volumetric images of iliac artery vasodilation over time in response to a bolus injection of sildenafil. The artery increased in diameter at 2 min post-injection and returned to its original diameter at 18 min post-injection. Figure 2b shows the acute vasodilation of the internal thoracic artery over time in response to a bolus injection of G-1. The artery has a peak vasodilatory response to G-1 at 18 min post injection, which is longer than sildenafil. We compared the maximum increase in the diameter of the major arteries for all treatment and control groups (Fig. 2c). The major vessels of the sildenafil and G-1 treated groups had a maximum vasodilation of 12% and 16% respectively, representing a significant increase in diameter ($p < 0.05$) compared to PBS treated controls for both cases, as assessed via one-way ANOVA followed by ad hoc $t$-test. Figure 2d shows the vessel response times to sildenafil and G-1. Arteries respond to sildenafil (~4 min) much faster than G-1 (~19 min).

To validate the measurements of vasodilation acquired using photoacoustic tomography, we monitored acute vasodilation of a

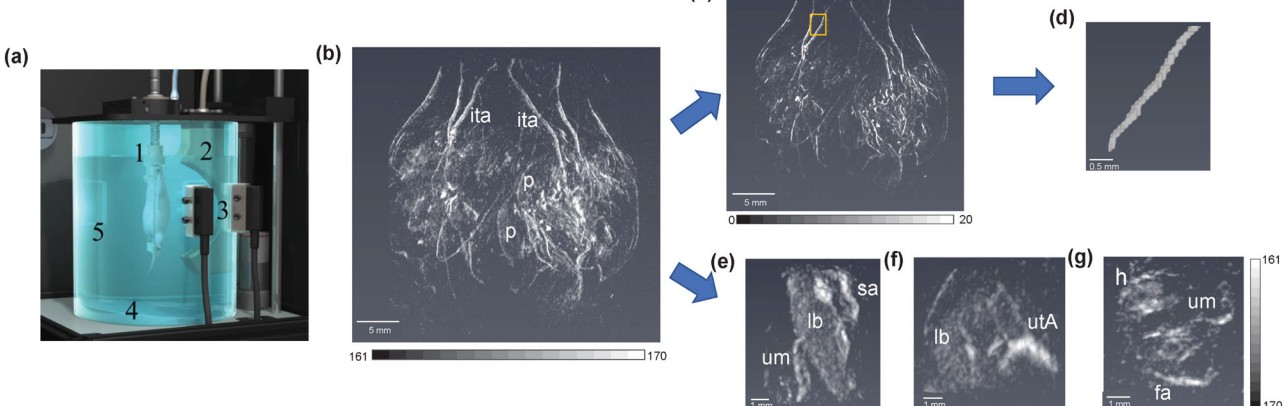

**Fig. 1 | Photoacoustic tomography of systemic vasculature of in vivo mouse.** **a** Imaging chamber of TriTom system. **1** Animal holder with nose cone, (**2**) Arc array detector, (**3**) Excitation termini, (**4**) Heating pad, (**5**) Water tank. **b** Representative 3D volume of a pregnant mouse indicating internal thoracic artery (ita) and placentas (p), (**c**) 3D volume filtered with frangi vesselness filter, (**d**) 3D volume of a section of artery indicated with yellow box in (**c**). **e** A single placenta with spiral artery (sa), labyrinth (lb), and umbilical cord (um). **f** Uterine artery (utA) feeding the placenta. **g** 3D volume of fetus indicating head (**h**), umbilical cord (um) and fetal aorta (fa).

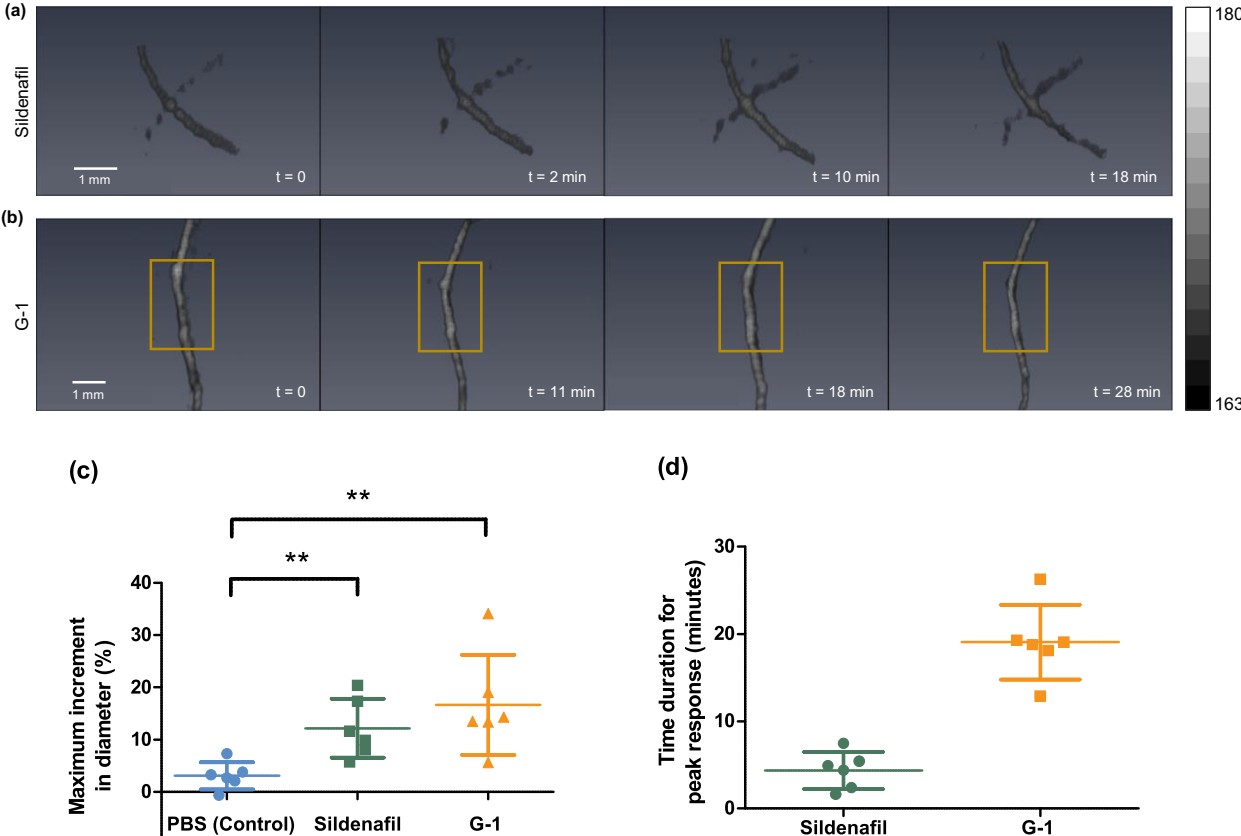

**Fig. 2 | Acute vasodilation of systemic vasculature. a** Representative 3D volumes of an iliac artery in response to sildenafil over time. **b** Representative 3D volumes of an internal thoracic artery in response to G-1 over time. **c** Scatter plot of maximum increase in vessel diameter from pre-injection diameter with mean and standard deviation ($n = 6$ mice/group). Sildenafil and G−1 treated groups showed a significant increase compared to PBS (control) with a $p$-value of 0.0086 (**) assessed via one-way ANOVA followed by ad hoc $t$-test (two-tailed, G-1 vs. PBS $p$-value 0.0073(**), Sildenafil vs. PBS $p$-value 0.0049(**)). **d** Scatter plot of time to peak response of sildenafil and G-1 with mean and standard deviation ($n = 6$ mice/group). The color bar represents the normalized photoacoustic signal intensity. Source data are provided as a Source Data file.

single artery using B-mode ultrasound imaging (Vevo 2100, FUJIFILM VisualSonics, Inc., Toronto, Canada). Since G-1 was previously reported to vasodilate many different arterial beds, we used this treatment to validate the extent of acute vasodilation and time of response using ultrasound imaging. The vasodilation peak response time measured with B-mode ultrasound imaging was the same as that measured with photoacoustic tomography imaging (Fig. 3e). However, the extent of the vasodilation observed in B-mode imaging was higher than the vasodilation observed in photoacoustic tomography (Fig. 3d). We assessed the accuracy of the TriTom photoacoustic tomography system to measure vessel diameters using a vessel-mimicking tube phantom (detailed in supplementary S.2.). Our tube phantom experiment showed a similar discrepancy in dilation measurement while comparing the increase in diameter from the camera image and photoacoustic image.

### Effects of selective vasodilators on maternal-fetal vasculature

We next measured the effect of the vasodilators on the uterine artery and fetal vasculature. Figure 4a and d show 3D volume images of the uterine artery feeding the placenta and the fetus, respectively. Due to the convoluted nature of uterine arteries, we measured the photoacoustic signal intensity as a measure of blood volume instead of vessel diameter. Both sildenafil and G-1 increased the photoacoustic signal intensity of the uterine artery, indicating vasodilation. We compared the increase in photoacoustic signal intensity of uterine artery in response to PBS, sildenafil and G-1 (Fig. 4b). There was a significant increase in the photoacoustic signal intensity of the uterine

artery ($p < 0.05$) compared with PBS. Our results align with previous studies demonstrating vasodilation of uterine arteries in response to sildenafil[41] and G-1[36] using single-vessel myography. In addition, we measured the photoacoustic signal intensity of fetal vasculature (Fig. 4d). Fetal photoacoustic signal intensity increased in response to both sildenafil and G-1 (Fig. 4e), however this increase was not significant.

### Effects of selective vasodilators on complex organ vessels

To demonstrate the effect of the vasodilators on the complex vasculature of an organ, the photoacoustic signal intensity of the placenta was monitored over time. An increase in placental photoacoustic signal intensity due to PBS, sildenafil and G-1 at 30 min post-injection was measured (Fig. 5c). In the fetal and placental tissues, the observed photoacoustic signal increase is due to increased blood volume in the vasculature which increases the tissue hemoglobin concentration and optical absorption. No increase in overall placental photoacoustic signal intensity over the 30 min imaging period was detected, indicating that total organ blood volume did not change in response to treatment. Maternal vessels supplying the placental labyrinth, the site of maternal-fetal oxygen and nutrient exchange, go through extensive remodeling during pregnancy, the spiral arteries losing smooth muscle cells to create low-flow blood niches for optimal maternal-fetal exchange[42,43]. Additionally, the fetal capillary network in the placental labyrinth is amuscular and therefore unresponsive to vasodilators[43]. Consistent with our in vivo findings, ex vivo human placentas have minimal arterial

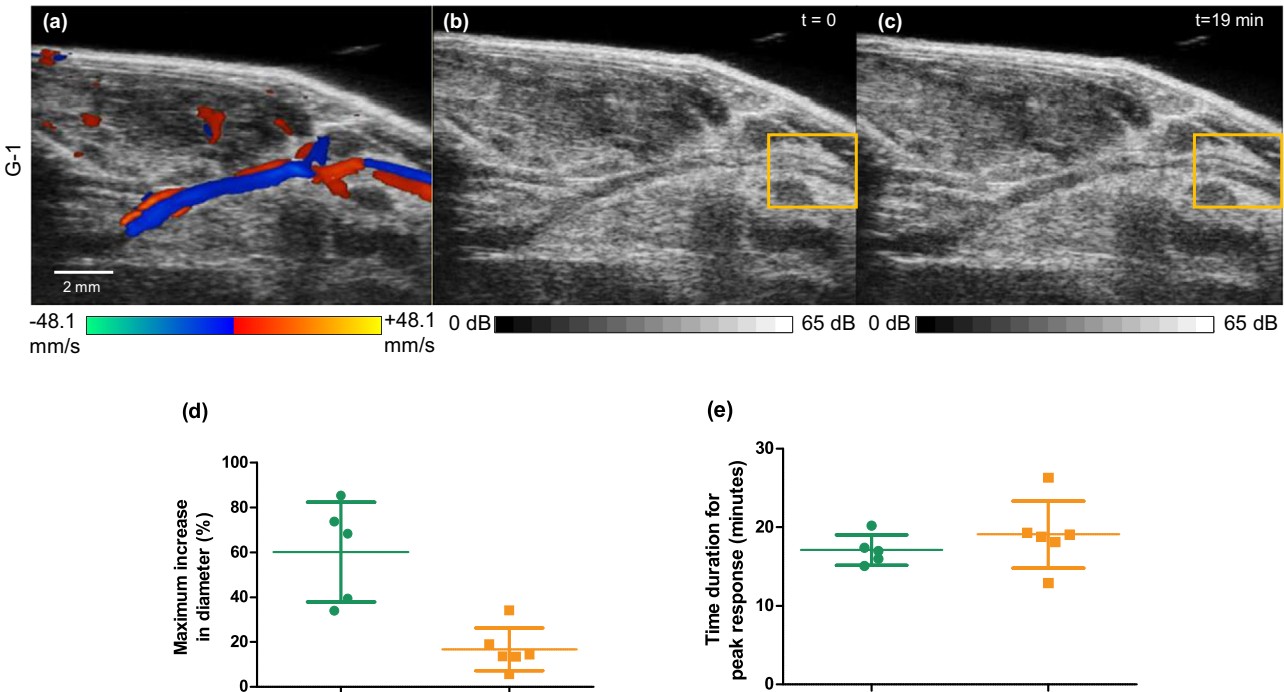

**Fig. 3 | Representative B-mode US imaging of a single artery. a** color Doppler image, (**b**) B-mode ultrasound image at $t = 0$. **c** B-mode ultrasound image at $t = 19$ min. The yellow box is indicating the artery monitored for vasodilation measurement. **d** Scatter plot of maximum increase in diameter using B-mode ultrasound and photoacoustic imaging with mean and standard deviation (B-mode US group: $n = 5$ mice, Photoacoustic group: $n = 6$ mice). **e** Scatter plot of peak time response for B-mode and photoacoustic acute vasodilation estimate with mean and standard deviation (B-mode US group: $n = 5$ mice, Photoacoustic group: $n = 6$ mice). Source data are provided as a Source Data file.

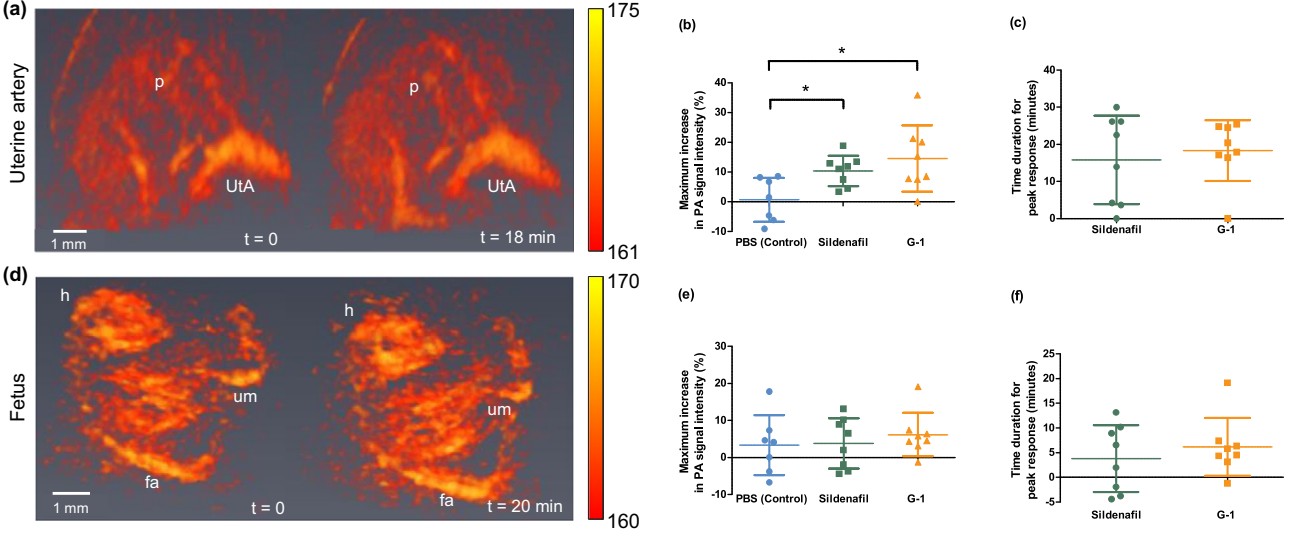

**Fig. 4 | 3D volumetric photoacoustic imaging of uterine and fetal responses to vasodilators. a** uterine artery and (**d**) fetal vasculature at different time points post-injection. Here, placenta (p), uterine artery (UtA), fetal head (h), umbilical cord (um) and fetal aorta (fa). **b–e** Scatter plot of maximum increase in photoacoustic signal intensity of uterine artery and fetal vasculature as a measure of blood volume from pre-injection state respectively with mean and standard deviation (G-1, Sildenafil: $n = 8$ mice/group, PBS: $n = 7$ mice). A significant increase in the photoacoustic signal intensity of the uterine artery compared with PBS with a $p$-value of 0.0136 (*) was tested via one-way ANOVA followed by an ad hoc $t$-test (two-tailed, G-1 vs. PBS $p$-value 0.0156(*), Sildenafil vs. PBS $p$-value 0.0103(*)). **c–f** Time duration for peak response of uterine artery and fetal vasculature due to sildenafil and G-1 respectively with mean and standard deviation ($n = 8$ mice/group). The color bar represents the normalized photoacoustic signal intensity. Source data are provided as a Source Data file.

vasoactivity within the placental labyrinth[44]. We also monitored the placental oxygenation in response to sildenafil (detailed in Supplement). However, we did not find a significant increase in placental oxygenation at 30 min post injection compared with PBS (Figure S.3).

## Discussion

We demonstrate spherical-view photoacoustic tomography (TriTom) for the quantification of in vivo vasodilation in multiscale vasculature during pregnancy. Our results showed that these techniques can monitor selective and acute vasodilation within a large abdominal area.

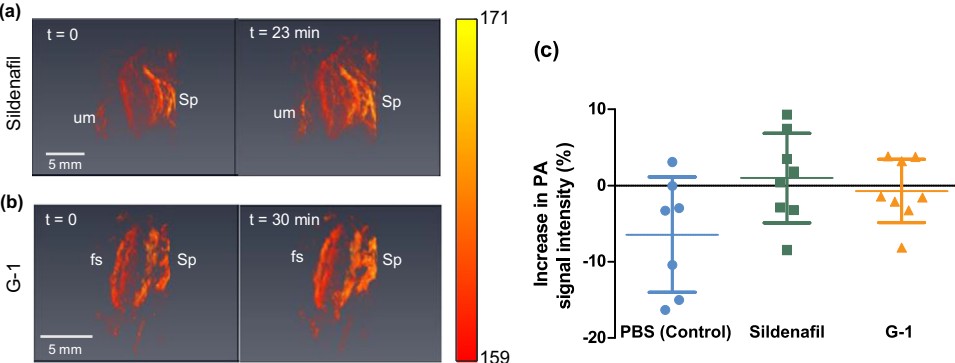

**Fig. 5 | 3D volumetric photoacoustic imaging placental response to vasodilators.** Representative 3D volumetric image of placenta indicating umbilical cord (um), spiral artery (Sp), and fetal side of the placenta (fs) in response to sildenafil (**a**) and G-1 (**b**) at different time points. **c** Scatter plot of increase in placental photoacoustic (PA) signal intensity as a measure of blood volume from pre- injection volume at 30 min post-injection with mean and standard deviation (G-1, Sildenafil: $n = 8$ mice/group, PBS: $n = 7$ mice). The color bar represents the normalized photoacoustic signal intensity. Source data are provided as a Source Data file.

We specifically quantified vasodilation in maternal arteries, uterine arteries supplying the placenta, fetal vasculature, and the placenta. The non-invasive tomography imaging system eliminates the single-vessel and ex vivo limitations of conventional myography techniques. In addition to in vivo systemic vasoactivity measurements, we found that the artery and placenta have different responses to two vasodilatory drugs, G-1 and sildenafil, when administered intravenously. The response time of the artery was validated by using standard B-Mode ultrasound imaging.

Photoacoustic tomography captured the vasoactivity of systemic arteries by imaging the extent of vasodilation of the arteries over time. Whereas the traditional B-mode ultrasound imaging captured the 2D cross-sectional image of vasodilation of a single artery, the TriTom system enabled us to measure vasodilation of multiple arteries simultaneously in 3D volume. In addition, vasodilation measurement from the 2D image may suffer from inaccuracy in selecting the mid-cross-section of the vessel, anatomical restriction, and differences in measurement due to selective vasoactivity in a different region of the body. TriTom system eliminates these difficulties by capturing the whole abdomen of the animal. Immersion in water could affect the autoregulation of blood flow and result in an increase in systemic blood flow[45]. However, the water pressure at the depth the mouse was immersed would be ~1.0068 atm. With this shallow water level of immersion, we do not expect an increase in cardiac output due to the water immersion.

We used two vasodilators with different mechanistic pathways to evaluate the ability of photoacoustic tomography to monitor differential responses in arteries. Both vasodilators showed significant vasodilation compared to the control treatment. For vasoactivity measurements, we selected internal thoracic arteries, iliac arteries, and superficial external pudendal artery (EPA). G-1 showed vasodilation in all or most arteries, whereas sildenafil mostly showed vasodilation of iliac and superficial EPA arteries (Supplementary Fig S.1). The measurement of vasodilation of these deep arteries did exhibit variability due to both biological and systemic effects. The artery location relative to the center of the arc array detector, the depth of the artery from the surface of the skin, and the variation in abdominal size can all introduce errors in the diameter estimates. As the expression of PDE5 is mostly found in the pelvic region[46], the iliac and superficial EPA arteries may have selective responses to sildenafil due to their location in the pelvic region. This suggests the selectivity of these vasodilators. Selective vasoactivity is important in determining the therapeutic responses of targeted vasculature in normal and diseased conditions as well as determining the efficacy of the therapeutics to relax the blood vessels[47]. The PAT methods developed here can be used to

evaluate the selectivity of vasodilators in vivo to optimize the targeted delivery of therapeutics and minimize off-target side effects. We performed pilot studies to evaluate the systemic arterial response to the vasoconstrictor endothelin-1 (supplementary S.3); however, vasoconstriction may reduce vessel diameters to sizes smaller than the spatial resolution of the imaging system. System improvements, such as increasing laser fluence, increasing image acquisition speed, and increasing the sensitivity and frequency of the transducer, would Increase sensitivity and/or spatial resolution to enable the measurement of vasoconstriction in future studies.

Although the time response for peak vasodilation is the same using ultrasound and photoacoustic imaging techniques, we observed a difference in the extent of vasodilation. The discrepancy in diameter estimation between B-mode ultrasound and photoacoustic imaging, as well as between camera images and photoacoustic images, is due to the uneven fluence distribution, back projection reconstruction, and the effects of the image acquisition time (essentially averaging the dynamic vessel response over 38 s, due to the lower temporal resolution of the PA system vs B-mode ultrasound)[48,49]. To estimate vessel diameters, we applied the Frangi vesselness filter, which also has limitations due to variation in depth, contrast, and a limited detection view for photoacoustic imaging[50]. However, we applied the filter to large and uniform vessels so that the filter can more accurately enhance the contrast. We also carefully selected filter parameters such as filter scale to ensure improvement in vessel contrast. The discrepancy in the extent of vasodilation may impact the sensitivity of the measurement of vasoactivity at low vasodilator doses. This impact may be mitigated through future imaging system enhancements such as increased laser fluence, higher sensitivity transducers, and more accurate reconstruction methods. To detect the vasodilation of less responsive arteries, increasing the vasodilator dose may be necessary[51]. B-mode ultrasound likely provides the most accurate vessel diameter estimation of available in vivo preclinical measurement tools. However, there may be variability in diameter estimation using B-mode ultrasound due to the human interface, scanning technology and/or biological factors[52,53].

Our imaging method measured the acute vasodilation of uterine arteries in response to vasodilators. Both vasodilators showed a significant increase in photoacoustic signal intensity of uterine arteries compared to our control treatment group. During pregnancy, vasodilation in the uterine artery plays a key role in maintaining uteroplacental blood flow and fetal growth[54]. Using our developed method, researchers can determine any alteration of vascular response in the uterine artery to identify complications such as fetal growth restriction, and preeclampsia. In addition to monitoring uterine

arteries, photoacoustic tomography monitored the effects of vasodilators on fetal vasculature. We did not observe any significant increase in photoacoustic signal intensity of fetal vasculature among groups. Previous studies showed sildenafil crosses the placental barrier using the human placenta perfusion model[55] and it improves fetal weight in the fetal growth restriction mouse model[56]. However, the acute administration of sildenafil and G-1 in our study may not affect the fetal vasculature during the observed time due to the low availability of drugs in fetal circulation and drugs crossing the barrier may be metabolized by the fetal liver before entering the fetal major vessels[57]. The ability of the TriTom system to observe fetal vasculatures provides the opportunity to monitor fetal response to therapeutics in longitudinal studies for chronic treatments.

We monitored the effects of vasodilators on the placenta by measuring the change in placental photoacoustic signal intensity as a measure of blood volume. Photoacoustic tomography captured the whole 3D volume of multiple placentas and enabled us to simultaneously monitor placental response to vasodilators. Due to the complexity of the placental vascular network, we monitored the vasoactivity of the whole placenta rather than measuring the vasodilation of individual vessels. In our study, we did not observe any measurable increase in photoacoustic signal intensity of the placenta. Other studies on the human placenta demonstrated the vasoactivity of myometrial and chorionic plate arteries of the placenta to vasodilators[44,58]. However, these studies did not show the in vivo effects of vasodilators on placental perfusion. Photoacoustic tomography allows studying true in vivo placental function in small animals which may not be possible with ex vivo vasoactivity measurement techniques. In supplementary S.4., we demonstrate the effects of sildenafil on placental oxygenation using the wavelength-dependent optical absorption properties of hemoglobin. Consistent with the effects of vasodilators on placental blood volume, we did not find any significant increase in placental oxygenation compared with the control group (Fig.S.4). Other imaging modalities captured the blood perfusion of the placenta using Doppler ultrasound, contrast-enhanced ultrasound, or magnetic resonance imaging (MRI) clinically[59–61]. However, these methods indirectly assessed the placental function. For example, power Doppler ultrasound measures the vascular indices of the placenta through blood flow measurement to estimate placental function. MRI uses a diffusion-weighted sequence that measures the microscopic diffusion properties of water to estimate the perfusion fraction of the placenta that alters during a diseased condition. In contrast, photoacoustic tomography captured the change in placental oxygenation that provided a more direct measurement of functional response. Our group previously demonstrated that 2D photoacoustic imaging showed an increase in placental oxygenation due to chronic treatment of sildenafil in a preeclamptic rat model[30]. Using the method, we applied in this work; we can capture the acute functional response of the placenta in vivo non-invasively.

In conclusion, we demonstrated photoacoustic tomography to assess in vivo systemic acute vasoactivity of small animals. Photoacoustic tomography enabled us to visualize the 3D volume of systemic arteries of maternal vasculature, placentas, and fetal vasculature. The large field of view of the imaging system coupled with fast scanning speed provided the opportunity to study dynamic systemic vasoactivity in vivo which is not possible with conventional myography techniques. We showed promising results on noninvasively measuring the vasoactivity of systemic arteries in response to vasodilators. We demonstrated the effects of vasodilators on the uterine artery and fetal vasculature as well as the placenta. In addition, we used the developed methods to assess placental oxygenation in response to acute vasodilation of the uterine arteries; while the therapeutics tested here were selected due to their potential to increase placental oxygenation, an increase in acute placental oxygenation was not detected. The developed methods therefore may be impactful for identifying improved

therapies for preeclampsia, which is characterized by placental ischemia. Photoacoustic tomography can contribute to the field of vascular physiology by providing information on the effects of therapeutics on the systemic vasculature. In addition, this technology can be applied to test novel therapeutics on animal models to study their systemic and off-target side effects. This technology could be applied to the study of cardiovascular metabolic diseases such as hypertension, diabetes, peripheral arterial disease as well as fetal growth restriction, preeclampsia, and gestational diabetes mellitus.

## Methods

### Animals
Tulane University maintains animal facilities accredited by the Association for Assessment and Accreditation of Laboratory Animal Care (AAALAC). All studies followed Protocol 1586, approved by the Tulane University Institutional Animal Care and Use Committee (IACUC). Pregnant CD-1 mice (Charles River, female timed pregnant, 8–10 weeks of age) at gestational day 16 were used for the described studies. Mice were housed on hardwood maple bedding (no. 7090, Sanichips, Harlan Teklad, Madison, WI; bedding changes once every 2 weeks) and shredded paper nesting material (Bed-r'Nest, The Andersons, Maumee, OH) in NexGen Mouse 500 cages. Mice had ad libitum access to acidified tap water and rodent chow (5053 Irradiated Laboratory Rodent Diet, Purina, Richmond, IN). Mice were maintained on a 12:12 light-dark cycle. Temperature (20–26 °C; 68–79 °F) and humidity (30–70%) were maintained. Only female pregnant animals were used in this study consistent with our overarching research program studying the specific vascular responses of the placenta.

The animals were anesthetized using isoflurane (1–3%) mixed with oxygen gas (1–2 L/min). Following induction of anesthesia, the fur from the torso of the animal was removed by shaving and application of depilatory cream. A PBS-filled catheter was then surgically placed in the right jugular vein under aseptic conditions for the administration of vasodilators during imaging. After the jugular vein catheter placement, the animals were transferred to the TriTom for imaging. The imaging chamber was filled with deionized and degassed water at 36 °C.

### Drugs
We treated the animals with the vasodilators sildenafil (sildenafil citrate, PHR1807, Sigma-Aldrich) and the G protein-coupled receptor G-1 (100089335, Cayman Chemical, Inc.). A volume of 0.1 mL of either G-1 (100 µg/kg of body weight in 5% DMSO and PBS), sildenafil (1 mg/kg body weight in PBS) or control PBS was administered through the right jugular vein catheter, followed by a flush of 0.1 mL PBS, while imaging continuously.

### Validation of time course using ultrasound imaging
B-mode ultrasound imaging (Vevo 2100, FUJIFILM VisualSonics, Inc., Toronto) was used to validate the extent of vasodilation and response time, using a single artery and G-1. The iliac artery ($n = 5$) branching from the abdominal aorta, was first identified using color Doppler ultrasound imaging (Fig. 3). The cross-sectional change in diameter of the artery was monitored continuously before, during and post drug administration for 1 h. The diameter of the artery was measured manually using the measurement tools in the VevoLab software.

### Spherical-view photoacoustic tomography system
A commercial photoacoustic/fluorescence tomography system TriTom (PhotoSound Technologies, Inc., Houston, TX, USA) was used for all image acquisitions. Figure 1a shows a schematic diagram of the TriTom system. The TriTom system uses a 6 MHz central frequency arc transducer array with 96 channels for photoacoustic signal acquisition. An Opotek Phocus Benchtop laser (5–7 ns pulses, 10 Hz repetition rate; Opotek Inc., Carlsbad, CA, USA) was integrated with the TriTom system for photoacoustic signal generation. The schematic diagram of

TriTom system is shown in Fig. 1a. To illuminate the object of interest, four optical fiber terminals mounted on the water tank of the imaging chamber at 45 and 90 degrees with respect to the vertical plane of the arc array transducer. A stepper motor is attached to the nose cone to rotate the object of interest mechanically in 360° while imaging. This nose cone delivers the anesthetic gas to the animal under water via anesthesia tube connected to animal holder. Deionized and degassed water at 36 °C temperature is used as acoustic coupling medium. The TriTom system spatial resolution in the transverse plane is $173 \pm 17$ μm and $640 \pm 120$ μm in the longitudinal axis with an absorption coefficient sensitivity detection limit of 0.258 cm$^{-1}$, and imaging speed of 38 s[22].

### In vivo photoacoustic imaging of vasoactivity

To assess the acute vascular reactivity in systemic and placental vasculature, three cohorts of animals for G-1 ($n = 8$), sildenafil ($n = 8$) and control ($n = 8$) were randomly assigned for this experiment. The Tri-Tom photoacoustic tomography system was used to acquire images. The 3D images were acquired before, during and after drug administration at 808 nm, the isosbestic point of the optical absorption of hemoglobin and oxyhemoglobin for 30 min. The laser fluence at the surface of the mouse skin was calculated to be 0.27, 0.325 and 0.142 mJ/cm$^2$ for 690, 808 and 890 nm wavelengths respectively. The photoacoustic images were reconstructed using a standard modified back-projection algorithm (TriTom reconstruction software V3.0.3)[40]. First, we reconstructed a 2D cross sectional photoacoustic image of the segmented 3D volume at $z = 0$ mm with varying speed of sound. Then the speed of sound which enhances the signal of the anatomical marker (e.g. cross section of an artery or placenta) appropriately was selected for each animal dataset. After reconstruction, additional processing steps were performed in Matlab V2021b. We normalized the reconstructed images by the laser energy during imaging and the average radio frequency (RF) signal data to minimize the variation between experimental groups. The average RF signal was calculated from the envelop of RF signal of each frame per channel and averaged over all channels. Then we rescaled all the normalized reconstructed images using the minimum and maximum intensity value for whole dataset to set same color scale from 0 to 2$^{16}$-1. We imaged the whole abdomen of a pregnant mouse in vivo capturing the systemic and fetal vasculature and placentas as shown in Fig. 1b. In this 3D volumetric image, we monitored the systemic arteries including the internal thoracic artery, iliac artery branching from the abdominal aorta as well as uterine artery. In addition, we monitored multiple placentas with their fetuses where spiral arteries feeding the placenta and umbilical cord connecting the fetal side of the placenta to the fetus (Fig. 1c).

To measure the diameter of the artery, we applied a Frangi vesselness filter to enhance the vascular signal in the 3D images[62]. The 3D images were visualized in Amira V6.0.1 (Thermo Fisher Scientific, Waltham, MA). We manually cropped the 3D volume to include internal thoracic arteries, iliac artery, and superficial EPA arteries identified in the full 3D image. First, we cropped each 3D image to a $3.5 \times 3.5 \times 3.5$ mm field of view containing each artery. The area of the artery within each 2D slice of the field of view was defined using image processing and segmentation tools in Amira. From these areas, the diameter of the artery in each 2D slice was calculated. To remove noise, we applied a moving average filter to the calculated diameter per slice with a window size of 10 pixels (1 mm) over the 3D field of view. The maximum averaged diameter from each segmented volume was used in the compiled data.

The maximum increase of arterial diameter($D$) from pre-injection time point was measured using the following equation,

$$Increment(\%) = \frac{D_t - D_{t0}}{D_{t0}} \times 100\% \qquad (1)$$

Where, $D_{t0}$ is the pre-injection estimated averaged diameter and $D_t$ is the estimated diameter at $t$ time point.

We compared the vasodilation of individual arteries due to sildenafil and G-1. We selected internal thoracic arteries, iliac artery, and superficial EPA from each animal to measure the vasodilation. Fig. S.1. shows the vasodilation of each artery due to sildenafil and G-1. We measured vasodilation due to sildenafil in iliac and superficial EPA arteries but not in internal thoracic arteries. In contrast, G-1 showed vasodilation in all three arteries. Since oxyhemoglobin absorbs less light at 690 nm than 850 nm, arteries were distinguished from veins by comparing the photoacoustic signal at these two wavelengths; vessels with high photoacoustic signal at 850 nm were identified as arteries. After identification of arteries at 850 nm, the laser was switched to 808 nm to track arterial diameters without interference from oxygenation. The maximum increase of arterial diameter was calculated as the difference from the pre-injection state of the diameter.

To measure the photoacoustic signal intensity of placenta, fetal vasculature, and uterine artery volumes, we manually segmented the 3D volumes from normalized reconstructed 3D images using MATLAB. Then we selected a noise intensity threshold for placenta, fetus, and uterine artery volumes separately to remove background noise from 3D volumes manually. The noise thresholds for the placenta, fetus and uterine arteries were then applied for entire dataset of placentas, fetuses, and uterine arteries. After applying the noise threshold, we measured the summation of photoacoustic signal intensity of the segmented volume for each time point. For each animal, the estimated total photoacoustic signal intensity of placentas (2 to 3 placentas), fetuses (2–3 fetuses) and uterine artery (1–2 sections of artery) at each time point were averaged. The maximum increment of total photoacoustic signal intensity of each volume (V) from pre-injection time point was measured using the following equation,

$$Increment(\%) = \frac{V_t - V_{t0}}{V_{t0}} \times 100\% \qquad (2)$$

Where, $V_{t0}$ is the pre-injection estimated total photoacoustic signal intensity and $V_t$ is the estimated total photoacoustic signal intensity at $t$ time point. The percent maximum increase in photoacoustic signal intensity was calculated as the difference between each time point and the pre-injection photoacoustic signal intensity.

### Statistical analysis

All data are expressed as mean ± standard deviation for each cohort. The measurements were performed on distinct samples. A priori power analysis was performed with a power of 0.80 and significance of 0.05 to determine the sample size using G*Power software (Heinrich-Heine-Universität, Dusseldorf, Germany). GraphPad Prism 5.0 (San Diego, CA) was used for all post hoc analysis. One way ANOVA with a $p$-value of 0.05 was performed to investigate the significance of the maximum increase of estimated diameter of arteries or estimated total photoacoustic signal intensity (placenta, fetus, or uterine artery) among three cohorts. A two-tailed $t$-test was also performed to investigate the significance of control (PBS) vs. sildenafil and control (PBS) vs. G-1 groups.

### Reporting summary

Further information on research design is available in the Nature Portfolio Reporting Summary linked to this article.

## Data availability

All image data are available on Dryad (https://doi.org/10.5061/dryad.sn02v6x9n)[63]. Source data for all figures are provided with this paper as a supplementary Source_data.xlsx. Source data are provided with this paper.

## Code availability

PhotoSound Image Reconstruction V3.0.3, GraphPad Prism 5.0, G*Power 3.1.9.6 for Mac, Amira V6.0.1, and Matlab 2021b were used to process all data. The post-reconstruction image processing codes are available upon request; this code includes implementation of the Frangi vesselness filter and linear fitting of the multiwavelength photoacoustic data to the oxygen saturation of hemoglobin, both of which are previously published and documented methods.

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

## Acknowledgements

We would like to thank Refik Mert Cam of University Illinois Urbana-Champaign for his suggestions for tomographic image rendering. We would like to acknowledge our funding sources from National Institutes of Health (NIH) R01 HD097466 (C.L.B.), P01 AG071746 (S.H.L.) and R01 HL133619 (S.H.L.) and Research Corporation for Science Advancement, Chan-Zuckerberg Initiative (C.L.B).

## Author contributions

K.H. completed all in vivo imaging experiments, data analysis and wrote the manuscript with support from C.L.B. D.J.L. contributed to the conception of the experiments. W.T. contributed to the experimental design of the phantom validation experiments. S.H.L. contributed to the experimental concept and assessment of vasodilation. C.L.B. conceived of and supervised the experimental design and analysis. All authors contributed to interpretation of the results and review of the final manuscript.

## Competing interests

The authors declare the following competing interests: The TriTom system (Photosound Technologies, Inc) was provided to the laboratory of C.L.B. for evaluation at no cost. D.J.L. and W.T. are currently employees of Photosound Technologies, Inc and W.T. has equity in the company; D.J.L.'s involvement in the design of the studies was completed while he was a trainee with C.L.B. and prior to his employment at Photosound Technologies. K.H. completed all experimental studies and primary data analysis and has no competing interests to declare. S.H.L. provided independent expertize in vasoactivity and experimental design and has no competing interests to declare.
