## [Peer Review File · Nature Communications]

In vivo noninvasive systemic myography of acute systemic vasoactivity in female pregnant miceREVIEWER COMMENTS

Reviewer #1 (Remarks to the Author):

This is an elegant study which highlights the use of spherical view photoacoustic tomography to noninvasively assess regional and temporal effects of vasoactive therapeutics. The findings will be of interest to the field, with biomedical and clinical relevance, however a number of issues remain to be addressed.

1. The photoacoustic method underestimates the extent of vasodilation. Can the authors comment on the limitations this may impose when assessing in vivo systemic vasoactivity and the therapeutic utility of vasodilators.
2. Fig S.1. There is a large degree of variability in the maximum increase in diameter in iliac and superficial EPA in response to both sildenafil and G-1. Can the authors provide an explanation for these observations.
3. The technology relies on the absorption of light by haemoglobin. How stable is the relative haemoglobin signal in the different vascular beds?
4. The study has focused on vasodilator agents. Using photoacoustic tomography, do the authors have evidence for selective actions of vasoconstrictors between the maternal systemic and fetal vasculature? Such findings would strengthen the study.
5. It is stated that an increase in placental oxygenation was observed at 30 minutes post injection in response to sildenafil (Page 169, Line 192). However, the increase in oxygenation in response to sildenafil did not differ as compared to that seen with PBS. As per the discussion, this statement in the results should be modified to indicate that sildenafil did not change placental oxygenation.
6. An increase in placental oxygenation was not observed with sildenafil. Can a positive control be used to demonstrate an increase in placental oxygenation under these experimental conditions?
7. Figure 3: Include in legend an indication that the acute vasodilation is in response to G-1.

8. Page 169, Line 183: Fig. 4c should read Fig. 5c

Reviewer #2 (Remarks to the Author):

The paper describes an application study using a commercial photoacoustic imaging product (TriTom from PhotoSound). Since some author is from that company, the conflict of interest should be clearly mentioned in the manuscript. Although the application study can be potentially interesting, there are many details to be addressed, and the novelty is not very clear. Here are some major concerns:

When comparing to ultrasound imaging, the author claims that standard ultrasound B-mode imaging can only acquire a 2D field of view. Although it is correct, the author may also want to compare it with ultrasound computed tomography, which can acquire a 3D field of view.

As for the comparison of previous photoacoustic imaging systems, the reference of (14) and (15) are not really relevant. Reference (14) is for photoacoustic microscopy, while reference (15) is for photoacoustic/optoacoustic mesoscopic. It's unfair to compare the proposed photoacoustic computed tomography (PACT) with those systems. There are plenty of PACT systems reported with better performances than the proposed system here, including spherical-view PACT. Please clarify the novelty or improvement of the proposed PACT system.

The system used in the research needs to be better characterized, such as spatial resolution and signal-to-noise ratio in 3D. Also, the experiment condition needs to be included, such as the laser fluence.

As shown in Fig. 1a, the mouse is fully immersed in the water bath. Will that water pressure affect physiology? Is the water temperature controlled and monitored to maintain animal body temperature?

The work focused on arteries instead of veins. How did the author differentiate arteries from veins? Looking at only the anatomy can be sometimes misleading, or even wrong.

What does the color bar number mean in all figures? It should be clarified.

How is the vessel diameter quantified from the 3D image? What threshold is determined to calculate the diameter? Especially, sometimes the vessel cross-section may not be round in shape.

The results in Fig. 2a and Fig. 2b show different time points, is there any specific reason? What is the time interval for different measurements? In addition, it is challenging to see the diameter changes directly from the image. It may be better to put close-up images to show the difference.

Figure 2d is not discussed or mentioned in the manuscript at all. What is the definition of time duration for peak responses?

Fig. 3d showed discrepancies between B-mode ultrasound and photoacoustic imaging, also observed between camera images and photoacoustic images. What is the cause of the discrepancy? Since the images went through Frangi vesselness filter, which is a nonlinear operator, it may overestimate or underestimate the vessel diameter. In addition, does the B-mode ultrasound image estimate the vessel diameter accurately?

What is the major cause of photoacoustic signal increase in Fig. 4 and Fig. 5? Higher hemoglobin concentration, changed oxygen saturation, or just bigger blood volume in the acoustical voxel?

The TriTom system 4 uses a 6 MHz central frequency arc transducer array with 96 channels for photoacoustic signal acquisition. Any reason did the authors pick up this frequency and channels?

RESPONSE TO REVIEWERS

We thank the reviewers for their careful consideration and review of our manuscript. We have addressed all comments, as detailed below, and we feel this revision is greatly improved.

Reviewer #1 (Remarks to the Author):

This is an elegant study which highlights the use of spherical view photoacoustic tomography to noninvasively assess regional and temporal effects of vasoactive therapeutics. The findings will be of interest to the field, with biomedical and clinical relevance, however a number of issues remain to be addressed.

1. The photoacoustic method underestimates the extent of vasodilation. Can the authors comment on the limitations this may impose when assessing in vivo systemic vasoactivity and the therapeutic utility of vasodilators.

We thank you for this question. We have added the following additional explanation of impacts and potential methods to mitigate these impacts to the Discussion:

“The discrepancy in the extent of vasodilation may impact the sensitivity of the vasoactivity measurement at low vasodilator doses. This impact may be mitigated through future imaging system enhancements such as increased laser fluence, higher sensitivity transducers, and more accurate reconstruction methods. To detect the vasodilation of less responsive arteries, increasing the vasodilator dose may be necessary.”

2. Fig S.1. There is a large degree of variability in the maximum increase in diameter in iliac and superficial EPA in response to both sildenafil and G-1. Can the authors provide an explanation for these observations.

We have added the following explanation of sources of variation in our imaging method to the Discussion:

“The measurement of vasodilation of these deep arteries did exhibit variability due to both biological and systemic effects. The artery location relative to the center of the arc array detector, the depth of the artery from the surface of the skin, and the variation in abdominal size can all introduce errors in the diameter estimates.”

3. The technology relies on the absorption of light by haemoglobin. How stable is the relative haemoglobin signal in the different vascular beds?

There are likely many factors which could influence the blood flow to different vascular beds; we've added discussion on several of these factors (please see response to Reviewer 2 comments number 5 and 12).

4. The study has focused on vasodilator agents. Using photoacoustic tomography, do the authors have evidence for selective actions of vasoconstrictors between the maternal systemic and fetal vasculature? Such findings would strengthen the study.

We have added the following to the Discussion:

“We performed pilot studies to evaluate the systemic arterial response to the vasoconstrictor endothelin-1; however, vasoconstriction may reduce vessel diameters to sizes smaller than the spatial resolution of the imaging system. Increasing system sensitivity and spatial resolution by increasing laser fluence, increasing image acquisition speed, and increasing the sensitivity of the transducer could enable the measurement of vasoconstriction in future studies.”

5. It is stated that an increase in placental oxygenation was observed at 30 minutes post injection in response to sildenafil (Page 169, Line 192). However, the increase in oxygenation in response to sildenafil did not differ as compared to that seen with PBS. As per the discussion, this statement in the results should be modified to indicate that sildenafil did not change placental oxygenation.

We apologize for this oversight. We have updated as follows,

“However, we did not find a significant increase in placental oxygenation at 30 minutes post injection compared with PBS (Figure S.3).”

6. An increase in placental oxygenation was not observed with sildenafil. Can a positive control be used to demonstrate an increase in placental oxygenation under these experimental conditions?

This is an area of future interest and study; our laboratory uses imaging tools to study preeclampsia, which is characterized by placental ischemia. Using these imaging methods to discover therapeutics which increase placental oxygenation is precisely the impact we hope these methods to have; at the moment, little is known about which therapeutic can acutely increase placental oxygenation. We have added the following to the Discussion to highlight this urgent application need:

“while the therapeutics tested here were selected due to their potential to increase placental oxygenation, an increase in acute placental oxygenation was not detected. The developed methods therefore may be impactful for identifying improved therapies for preeclampsia, which is characterized by placental ischemia.”

7. Figure 3: Include in legend an indication that the acute vasodilation is in response to G-1.

The legend “G-1” was added to Figure 3.

8. Page 169, Line 183: Fig. 4c should read Fig. 5c

We have updated these errors.

Reviewer #2 (Remarks to the Author):

The paper describes an application study using a commercial photoacoustic imaging product (TriTom from PhotoSound). Since some author is from that company, the conflict of interest should be clearly mentioned in the manuscript. Although the application study can be potentially interesting, there are many details to be addressed, and the novelty is not very clear. Here are some major concerns:

When comparing to ultrasound imaging, the author claims that standard ultrasound B-mode imaging can only acquire a 2D field of view. Although it is correct, the author may also want to compare it with ultrasound computed tomography, which can acquire a 3D field of view.

Thank you for your review and consideration.

1. We have added the conflict of interest to the manuscript.
2. We have added the following information about 3D ultrasound tomography in comparison to 3D photoacoustic imaging.

“3D ultrasound tomography has lower vascular contrast in comparison to photoacoustic imaging^{14, 15, 16.}”

3. As for the comparison of previous photoacoustic imaging systems, the reference of (14) and (15) are not really relevant. Reference (14) is for photoacoustic microscopy, while reference (15) is for photoacoustic/optoacoustic mesoscopic. It's unfair to compare the proposed photoacoustic computed tomography (PACT) with those systems. There are plenty of PACT systems reported with better performances than the proposed system here, including spherical-view PACT. Please clarify the novelty or improvement of the proposed PACT system.

Thank you for your comments regarding references 14-15 (now 17-18). These references are cited since they are foundational studies first reporting the use of photoacoustic imaging specifically to monitor vasoactivity; we have cited these studies to clarify the novelty of our methods in comparison to these prior studies of vasoactivity. We have added the following paragraph in the “Main” section to compare the system used here to the performance of other state-of-the art tomography systems.

“However, these studies used photoacoustic microscopy systems with slow temporal resolution, limited field of view and limited imaging depth. The spherical-view photoacoustic tomography system used here provides in vivo high resolution anatomical and functional visualization of small animals. Currently, a temporal resolution of 2-10 seconds is the fastest reported imaging speed for spherical-view geometry¹⁹. Though this system has a high temporal resolution, the spatial resolution of the system is 390 μm (lateral) and 370 μm (axial) which is larger than the diameter of mouse arteries²⁰. Another spherical-view system has a reported spatial resolution of 200 μm (lateral and axial) and large FOV (80 mm)²¹. However, the limited view aperture of the system requires a helical scanning scheme for whole-body imaging, reducing the temporal resolution to 45 seconds. The TriTom spherical-view photoacoustic tomography system uses a 360° rotation to capture a 3 cm³ volume with a spatial resolution of 173 \pm 17 μm (transverse plane) that can image vessel diameters down to ~180 μm in 38 seconds²².”

4. The system used in the research needs to be better characterized, such as spatial resolution and signal-to-noise ratio in 3D. Also, the experiment condition needs to be included, such as the laser fluence.

Thank you for your suggestion. The system is characterized in this recent publication which has been added to the methods and described²².

“The TriTom system performance was characterized in a recent publication []. The spatial resolution in the transverse plane was $173 \pm 17 \mu\text{m}$ and $640 \pm 120 \mu\text{m}$ in the longitudinal axis with an absorption coefficient sensitivity detection limit of 0.258 cm^{-1} .”

The laser fluence has been added to the Methods as follows,

“The laser fluence at the surface of the mouse skin was calculated to be 0.27, 0.325 and 0.142 mJ/cm^2 at 690, 808 and 890 nm wavelengths respectively.”

5. As shown in Fig. 1a, the mouse is fully immersed in the water bath. Will that water pressure affect physiology? Is the water temperature controlled and monitored to maintain animal body temperature?

We have added the following to the Discussion:

Immersion in water could affect the autoregulation of blood flow and result in an increase in systemic blood flow ⁴⁵. However, the water pressure at the depth the mouse was immersed would be $\sim 1.0068 \text{ atm}$. With this shallow water level of immersion, we do not expect an increase in cardiac output due to the water immersion.

And to the Methods:

“The imaging chamber was filled with deionized and degassed water at $36 \text{ }^\circ\text{C}$.”

6. The work focused on arteries instead of veins. How did the author differentiate arteries from veins? Looking at only the anatomy can be sometimes misleading, or even wrong.

We have clarified our method of identification of arteries by adding the following to our Methods:

“Since oxyhemoglobin absorbs less light at 690 nm than 850 nm, arteries were distinguished from veins by comparing the photoacoustic signal at these two wavelengths; vessels with high photoacoustic signal at 850 nm were identified as arteries.”

7. What does the color bar number mean in all figures? It should be clarified.

The following line has been added to figure caption of Figs.2,4 and 5.

“The color bar represents the normalized photoacoustic signal intensity.”

After reconstruction, we normalized the reconstructed images by the laser energy measured during imaging and the average radio frequency (RF) signal data to minimize the variation between experimental groups. The average RF signal was calculated from the envelop of RF signal of each frame per channel and averaged over all channels. Then we rescaled all the normalized reconstructed images using the minimum and maximum intensity value for whole dataset to set same color scale from 0 to 2^{16} .

8. How is the vessel diameter quantified from the 3D image? What threshold is determined to calculate the diameter? Especially, sometimes the vessel cross-section may not be round in shape.

We have added the following detailed methods to the Supplement:

First, we cropped each 3D image to a $3.5 \times 3.5 \times 3.5$ mm field of view containing each artery. The area of the artery within each 2D slice of the field of view was defined using image processing and segmentation tools in Amira. From these areas, the diameter of the artery in each 2D slice was calculated. To remove noise, we applied a moving average filter to the calculated diameter per slice with a window size of 10 pixels (1 mm) over the 3D field of view. The maximum averaged diameter from each segmented volume was used in the compiled data.

9. The results in Fig. 2a and Fig. 2b show different time points, is there any specific reason? What is the time interval for different measurements? In addition, it is challenging to see the diameter changes directly from the image. It may be better to put close-up images to show the difference.

Different time points are shown in Fig. 2a and Fig. 2b due to the difference in the vessel response times to sildenafil and G-1. We have added this to the Results. We have updated Fig2b to show the artery volumes more clearly (see response to question 8 as well).

“Arteries respond to sildenafil (~4 minutes) much faster than G-1 (~19 minutes).”

10. Figure 2d is not discussed or mentioned in the manuscript at all. What is the definition of time duration for peak responses?

We apologize for the oversight. We have added the following in the Results:

“Fig. 2d shows the vessel response times to sildenafil and G-1.”

11. Fig. 3d showed discrepancies between B-mode ultrasound and photoacoustic imaging, also observed between camera images and photoacoustic images. What is the cause of the discrepancy? Since the images went through Frangi vesselness filter, which is a nonlinear operator, it may overestimate or underestimate the vessel diameter. In addition, does the B-mode ultrasound image estimate the vessel diameter accurately?

We have added the following to the discussion:

“The discrepancy in diameter estimation between B-mode ultrasound and photoacoustic imaging, as well as between camera images and photoacoustic images, is due to the uneven fluence distribution, back projection reconstruction, and the effects of the image acquisition time (essentially averaging the dynamic vessel response over 38 seconds, due to the lower temporal resolution of the PA system vs B-mode ultrasound). To estimate vessel diameters, we applied the Frangi vesselness filter which also has limitations due to variation in depth, contrast, and a limited detection view for photoacoustic imaging⁵⁰. However, we applied the filter to large and uniform vessels so that the filter can more accurately enhance the contrast. We also carefully selected filter parameters such as filter scale to ensure improvement in vessel contrast. “

And

“B-mode ultrasound likely provides the most accurate vessel diameter estimation of available in vivo preclinical measurement tools. However, there may be variability in diameter estimation using B-mode ultrasound due to the human interface, scanning technology and/or biological factors^{53, 54}.

12. What is the major cause of photoacoustic signal increase in Fig. 4 and Fig. 5? Higher hemoglobin concentration, changed oxygen saturation, or just bigger blood volume in the acoustical voxel?

The following explanation has been added to the Results,

“In the fetal and placental tissues, the observed photoacoustic signal increase is due to increased blood volume in the vasculature which increases the tissue hemoglobin concentration and optical absorption.”

13. The TriTom system uses a 6 MHz central frequency arc transducer array with 96 channels for photoacoustic signal acquisition. Any reason did the authors pick up this frequency and channels?

Due to the limited bandwidth of the detector, there is a trade-off between the imaging field of view and spatial resolution. In the design of the TriTom system, 6 MHz frequency was chosen to enable whole-body imaging of small animals, while still providing sufficient spatial resolution for the major vessels and organs of the animal. The current system uses 96 channels to reduce the complexity in data acquisition system and optimize hardware cost.

REVIEWER COMMENTS

Reviewer #1 (Remarks to the Author):

The authors have addressed the majority of the concerns raised. However, further clarification is required with regard to the underlying cause of the increase in photoacoustic signal intensity in response to vasodilators.

The photoacoustic signal can arise from both oxyhaemoglobin (absorbs more light at 850 nm) and deoxyhaemoglobin and the authors have identified arteries as those in which there is a high photoacoustic signal at 850nm. However, vasodilators may lead to a decrease in arterial oxygen saturation of haemoglobin (Zhu et al., 2022 Light: Science & Applications). What impact would this change in arterial oxygen saturation of haemoglobin have upon the photoacoustic signal detected at 850nm? Is it conceivable that such a change in haemoglobin oxygen saturation, in response to a vasodilator, may attenuate the photoacoustic signal intensity, despite an increase in tissue haemoglobin concentration per se, and as such the extent of vasodilation underestimated?

Reviewer #2 (Remarks to the Author):

The author has revised the parts reviewer asked. It was suggested to accept it for publication.

RESPONSE TO REVIEWERS

We thank the reviewers for their careful consideration and review of our manuscript. We have addressed the remaining comment, as detailed below.

We have also reformatted the manuscript to fit the editorial guidelines by reordering the sections as required, adding an Author Contributions statement, and adding a title to each figure legend.

The authors have addressed the majority of the concerns raised. However, further clarification is required with regard to the underlying cause of the increase in photoacoustic signal intensity in response to vasodilators.

The photoacoustic signal can arise from both oxyhaemoglobin (absorbs more light at 850 nm) and deoxyhaemoglobin and the authors have identified arteries as those in which there is a high photoacoustic signal at 850nm. However, vasodilators may lead to a decrease in arterial oxygen saturation of haemoglobin (Zhu et al., 2022 Light: Science & Applications). What impact would this change in arterial oxygen saturation of haemoglobin have upon the photoacoustic signal detected at 850nm? Is it conceivable that such a change in haemoglobin oxygen saturation, in response to a vasodilator, may attenuate the photoacoustic signal intensity, despite an increase in tissue haemoglobin concentration per se, and as such the extent of vasodilation underestimated?

To clarify, we used 850 nm to identify vessels as arteries prior to injection of the vasodilator, and then monitored the arterial diameter using 808 nm, where the oxygenation of the blood would not impact the estimate of the diameter.

We have added to the methods to clarify:

“After identification of arteries at 850 nm, the laser was switched to 808 nm to track arterial diameters without interference from oxygenation.”

REVIEWERS' COMMENTS

Reviewer #1 (Remarks to the Author):

The authors have revised the manuscript and addressed adequately my concerns.